# Analytical Perspectives in the Study of Polyvalent Interactions of Free and Surface-Bound Oligonucleotides and Their Implications in Affinity Biosensing

**DOI:** 10.3390/ijms24010175

**Published:** 2022-12-22

**Authors:** Laura-Elena Gliga, Bogdan-Cezar Iacob, Sanda-Nastasia Moldovean, David A. Spivak, Andreea Elena Bodoki, Ede Bodoki, Radu Oprean

**Affiliations:** 1Analytical Chemistry Department, “Iuliu Hațieganu” University of Medicine and Pharmacy, 4, Louis Pasteur St., 400349 Cluj-Napoca, Romania; 2Faculty of Physics, Babeş-Bolyai University, 1, Kogălniceanu St., 400084 Cluj-Napoca, Romania; 3Department of Chemistry, Louisiana State University, Baton Rouge, LA 70803, USA; 4Inorganic Chemistry Department, “Iuliu Hațieganu” University of Medicine and Pharmacy, 12, Ion Creangă St., 400010 Cluj-Napoca, Romania

**Keywords:** polyvalent oligonucleotide interactions, surface bound oligonucleotides, oligonucleotide sequence complementarity, capillary gel electrophoresis, microcalorimetry, molecular dynamics, affinity biosensing, dual molecular recognition strategies

## Abstract

The high affinity and/or selectivity of oligonucleotide-mediated binding offers a myriad of therapeutical and analytical applications, whose rational design implies an accurate knowledge of the involved molecular mechanisms, concurring equilibrium processes and key affinity parameters. Oligonucleotide-functionalized gold surfaces or nanostructures are regularly employed analytical platforms for the development of label-free optical or electrochemical biosensors, and recently, novel detection platform designs have been increasingly considering the synergistic effect of polyvalent binding, involving the simultaneous interaction of two or several oligonucleotide strands. Considering the general lack of studies involving ternary single-stranded DNA (ssDNA) interactions, a complementary analytical workflow involving capillary gel electrophoretic (CGE) mobility shift assay, microcalorimetry and computational modeling has been deployed for the characterization of a series of free and surface-bound binary and ternary oligonucleotide interactions. As a proof of concept, the DNA analogue of MicroRNA 21 (miR21), a well-known oncogenic short MicroRNA (miRNA) sequence, has been chosen as a target molecule, simulating limiting-case scenarios involved in dual molecular recognition models exploited in affinity (bio)sensing. Novel data for the characterization of oligonucleotide interacting modules is revealed, offering a fast and complete mapping of the specific or non-specific, often competing, binary and ternary order interactions in dynamic equilibria, occurring between various free and metal surface-bound oligonucleotides.

## 1. Introduction

Oligonucleotide-functionalized gold surfaces or nanostructures, using single or multiple target recognition strategies, are frequently employed analytical platforms for the development of label-free optical or electrochemical biosensors [1,2,3,4,5,6]. However, the covalent binding of an oligonucleotide sequence to the metallic surface, often through the aid of a flexible oligonucleotide linker [7,8], may decrease the degrees of freedom of the involved strand and lead to unexpected change in the binding kinetics, affinity, and selectivity of the molecular recognition unit [9]. Furthermore, novel detection platform designs are increasingly considering the synergistic effect of polyvalent binding [10], involving the simultaneous interaction of two or several oligonucleotide strands [11]. Consequently, it becomes essential to have the most accurate knowledge of the key affinity parameters of these nucleotide sequences and the influence of critical variables in the given environment of use. Both kinetic and structural studies are essential to reveal the mechanism of probe–target interaction, as well as some key binding features (affinity, rate constant, specificity, ion dependence, the influence of buffer), that are able to integrate and modulate the molecular recognition and sensing capabilities encoded within oligonucleotides into various bioanalytical applications.

Different analytical methods are currently used to determine the affinity (constant of dissociation, K_d_ or association, K_a_) of an oligonucleotide sequence toward its cognate target, such as: filter binding assays [12], fluorescence anisotropy [13], equilibrium dialysis, surface plasmon resonance (SPR), nuclear magnetic resonance (NMR), differential scanning calorimetry, and electrodriven or chromatographic separations [14,15,16]. For most of these methods, the labeling of either the targeted analyte or the probe is vital [17], which in turn may impact the interaction between the two sequences. SPR-based platforms are widely used [18], as they provide real-time kinetic and binding information; however, upon the ligand’s surface immobilization, it may no longer maintain its native configuration. Impeding its freedom of movement may restrict its adaptive folding, and mass transport limitations may also occur [19,20]. Capillary electrophoresis (CE), widely used in proteomics, immunology, and drug development, provides a fast separation of minute amounts of samples, with a minimal use of reagents and solvents. As a high-throughput separation method, gel-facilitated sieving (CGE) has already been extensively employed for DNA analysis [21], among the landmark applications being DNA sequencing and sizing [22], synthesis of DNA aptamers and assessment of aptamer binding affinity [23]. Similarly, with the electrophoretic mobility shift assay (EMSA), upon the changes in the electrophoretic mobility of free and bound sequences, CE provides quantitative means to measure the binding affinity of DNA oligonucleotides toward their targets [24]. Up until now, CE has been applied for the study of binding interactions between aptamers and a wide range of target molecules (proteins, peptides, lipopolysaccharides, and drug molecules) [16], but little to no investigation of the binding mechanism between two or more single-stranded DNA (ssDNA) strands has been reported.

Considering the general lack of studies involving ternary ssDNA interactions, here, we report an extensive study of the interaction of ssDNA sequences in binary and ternary interacting systems (Figure 1). Therefore, a simple, fast, quantitative and label-free, CGE-based methodology was developed to provide insights on the interaction mechanism of two or more DNA or RNA oligonucleotide strands with various biomolecular implications and applications (e.g., biosensor development, drug delivery systems, single guide RNA (sgRNA) design, CRISPR/Cas9 genome editing, etc.), and to investigate the predictability of base pairing between various cognate target analogues. Furthermore, identifying the influence on the affinity and selectivity of these molecular recognition units upon their surface binding to metallic nanostructures (gold nanorods (GNRs)) recurrently used in the development of biosensing platforms [25] was also attempted. Aiming to provide a deeper understanding of the interaction mechanisms and to outline vital thermodynamic aspects, CGE studies were complemented and cross-validated by microcalorimetric data and theoretical molecular dynamics (MD) simulations (Figure 1). As a proof of concept, the more stable and cost-effective DNA analogue of MicroRNA 21 (miR21), a well-known oncogenic short MicroRNA (miRNA) sequence [26,27,28] (Figure 2A), composed of 22 nucleotides involved in tumor initiation, progression, invasion, and metastasis in several tumor types [29,30], has been chosen as the target molecule and named in the current study as ‘miR21′.

## 2. Results and Discussion

As previously stated, this study aimed to investigate in more depth the simultaneous interaction between three ssDNA sequences and the particular case where one of them is covalently attached at the surface of GNRs. The binding mechanism of ssDNAs is strongly correlated to the comprised nucleic acid sequences and their spatial conformations. However, their dynamic evolution and/or inconvenient structural changes might involve “inadequate affinities” and therefore a lower binding specificity [31]. As such, the working protocol was designed to provide relevant information regarding binary and tertiary interaction between free or surface-attached ssDNA oligonucleotides by both CGE and isothermal titration calorimetry (ITC) by observing the particularities of those concurrent polyvalent interactions (Figure 1). Different aspects that might influence the interaction, such as the thermal denaturation of ssDNA oligonucleotides or the presence of physiological ions (100 mM Na^+^, 0.01 mM Mg^2+^), were also considered. The study of molecular interactions in a ternary system also aimed to determine the influence of the metal surface of nanostructures (i.e., GNRs) upon oligonucleotide functionalization as well as on the affinity and hybridization event expected to occur between complementary ssDNA sequences. 

To establish which system allows a better recognition of the ssDNA miR21 sequence (considered the target), two complementary oligonucleotides, engaged in a dual molecular recognition model, A1 and A2, with A2 being free or covalently bound at the surface of GNRs (A2@GNRs) were investigated in various formats (Figure 1). As further variables to investigate, which are often encountered in oligonucleotide-based affinity biosensor development making use of dual molecular recognition principles, the A1 sequence incorporated a non-binding iSp18 (hexa-ethyleneglycol) spacer and a 5′-phosphate-linked Acrydite™ extension (Figure 2A) offering further single-point covalent anchoring commonly used in biosensing applications [32]. Furthermore, the GNRs surface bound A2 sequence (A2@GNRs) was linked with an oligonucleotide spacer of 15 thymines (Sp15T), providing the required flexibility and degree of freedom in its interaction with other ssDNA sequences. As reference for the surface bound ssDNA oligonucleotide, a free A2 sequence, incorporating an iSp18 spacer and the corresponding 5′-phosphate-linked Acrydite™ extension, was also employed (Figure 2A). To evaluate the binding selectivity of the dual molecular recognition complex (A1–miR21–A2), three other ssDNA sequences (miR21-spacer (miR21-sp), anti-miR21 and random) (Figure 2A) with varying degrees of complementarity toward A1 or A2 were also tested.

Initially, the influence of prior thermal denaturation of the ssDNA sequences on the process of hybridization has been tested. The electrophoretic and microcalorimetric assays confirmed similar hybridization events with or without previous thermal denaturation for both the binary (A1–miR21 or A2–miR21) and ternary (A2@GNRs–miR21–A1) systems (Appendix A, Section 2.2. Microcalorimetric Study). Likewise, the presence of Mg^2+^ ions seemed not to be critical for the hybridization process (Appendix A). Therefore, unless stated differently, all affinity experiments have been performed without thermal unfolding of the oligonucleotides and in the absence of Mg^2+^ ions.

### 2.1. CGE Analysis

#### 2.1.1. Binary and Ternary Complex Assessment

In a CGE analysis, DNA molecules will electrophoretically migrate through the porous gel matrix toward the anode due to their negatively charged phosphate backbone, mostly by a size-dependent impeded manner. Upon binding to the target oligonucleotide with varying degree of complementarity, the distinct electrophoretic mobility of the emerging double-stranded DNA (dsDNA) supramolecular complex leads to a shift in the migration time compared to that of the free ssDNA fragments. Under the given CGE conditions, the investigated oligonucleotides demonstrated specific migration times, warranting a simple peak assignment and a straightforward monitoring of the hybridization process (Figure 2A). However, the GNRs-bound A2 oligonucleotide (A2@GNRs) could not be detected in the recorded electropherograms as the bulky metal nanostructure is unlikely to move through the viscous running buffer.

As a first step, the emerging binary complex resulting from the incubation of the two non-surface bound oligonucleotides with the highest complementarity, A1 and its oligonucleotide target (miR21), has been investigated by electrophoretic analysis. The particularities of the binary interaction and the ideal combination ratio between the two studied sequences were monitored at a fixed concentration (2.22 µM) of A1 and increasing levels of miR21 (0–8.88 µM) while tracking the peak area of the emerging binary complex (hybridized dsDNA). The highest yield for the binary complex was achieved at a miR21:A1 molar ratio slightly higher than 1, which was correlated with the absence of free A1. Any subsequent raise in miR21 levels rendered the peak of the binary complex unchanged (Figure 3A,E).

To confirm the hybridization of the second complementary sequence (A2) with the target oligonucleotide, the interaction of the free A2 and miR21 has been studied similarly. Once again, and as expected, the recorded signal for the hybridized miR21–A2 complex reaches a plateau at a molar ratio of ~1 of the interacting oligonucleotides (Figure 3B).

Subsequently, the formation of a binary complex between miR21 and A2 covalently attached to the surface of GNRs (A2@GNRs) has been investigated. By using the same experimental conditions, the concentration of A2@GNRs was kept constant (2.22 µM) while increasing the miR21 concentration in the range of 0–8.88 µM. Surprisingly, all the miR21 added to the system was identified as free-moving (Figure 3F, Table 1).

The use of two ssDNA oligonucleotides, A1 and A2, complementary toward two different binding zones of the target miR21 (Figure 2B), aimed to exploit their concurrent affinity. As such, the formation of a ternary complex between the target miR21, free A1 and free A2 has been evaluated maintaining a specific order in the preparation of the ternary incubates (3.2 ssDNA oligonucleotides hybridization conditions). First, A2 was admixed with miR21, which was followed by the addition of A1. For a molar ratio slightly higher than 1 between miR21:A1 and miR21:A2, the peaks corresponding to A1 and A2 disappeared, and the maximum ternary complex formation has been detected (Figure 3C). As the concentration of miR21 is further increased, the excess of the miR21 promotes reapportioning of the ternary complex to the more stable binary A1 or A2–miR21 complexes. This was clearly seen for the miR21–A1 complex; however, the miR21–A2 complex could not be resolved from the ternary complex.

Subsequently, the assessment of the similar ternary hybridization complex using the gold surface bound A2 oligonucleotide (A2@GNRs) while inducing constraints in their degree of freedom was also evaluated (Figure 3D,G). As in the previous experimental settings, the concentration of both A1 and A2 was kept constant (2.22 µM) while the concentration of the target oligonucleotide (miR21) was increased between 0 and 8.88 µM. The highest yield in a hybridized complex with electrophoretic mobility was achieved at a higher than 1 molar ratio between miR21:A1, and expectedly, no A2@GNRs was detected.

Overall, for the binary system consisting of miR21–A1 and for the ternary system consisting of A2@GNRs–miR21–A1, the presence of a novel peak with a slightly higher migration time than miR21, corresponding to a hybridized dsDNA complex, was noticed. It is suspected that the hybridized free-moving complex detected for the ternary A2@GNRs–miR21–A1 incubate is mainly consisting of a miR21–A1 dsDNA complex. Furthermore, the CGE hybridization profiles of the binary (miR21–A1) and ternary (A2@GNRs–miR21–A1) systems demonstrate very similar trends in terms of the detected dsDNA strand (Figure 3A,D).

By following the levels of free miR21 detected by CGE in the aforementioned binary and ternary incubates (Table 1) and especially the ones for the highest excess (8.88 µM) of the target oligonucleotide, it would seem that in the case of the binary incubate of A2@GNRs–miR21, no double-stranded complex formation is occurring, practically all added miR21 in the system being detected in its free form. Subsequently, in the binary miR21–A1 incubate, nearly 82% of the total miR21 added to the system is involved in the hybridization process, since approximately 18% was detected in its free form. Using 8.88 µM miR21, one would expect the same fraction of excess (free) target oligonucleotide also in the case of the ternary system, but in fact, the detected levels are more than double in this case (3.80 μM miR21, ~43%).

In the view of the very high fraction of free miR21 detected for the A2@GNRs–miR21 incubate and by using the simplest approach of reasoning, one would consider that the non-specific adsorption of free oligonucleotides at the GNRs surface is insignificant and that the hybridization between A2 and miR21 is somehow hindered. Nevertheless, these statements would speak against current knowledge on the hybridization of ssDNA strands bound to various gold surfaces in the development of biosensing platforms (SPR sensors, electrochemical biosensors, etc. [33]), and they are conflicting with the observed microcalorimetric data (Section 2.2. Microcalorimetric Study). The data can be explained by the hypothesis that the non-specifically adsorbed oligonucleotides (i.e., miR21 and A1) on the surface of A2@GNRs [34] during the incubation period are stripped off during the electrophoretic sieving, noting that gel electrophoresis is commonly used for the isolation and clean-up of oligonucleotide-functionalized noble metal colloids from starting materials and side products [35]. Since all CGE separations were performed well under the estimated melting points (~45–50 °C) of the studied ssDNA oligonucleotides, that rules out the possibility of the thermodynamically driven dissociation of the oligonucleotide duplexes into ssDNA. Moreover, the hybridization of complementary oligonucleotides not covalently bound to GNRs (e.g., binary miR21–A1 or ternary A2–miR21–A1 complexes) is clearly detected in CGE using the same experimental conditions. Therefore, subjecting the ternary oligonucleotide incubate A2@GNRs–miR21–A1 to the CGE separation leads most probably to the detection of the same miR21–A1 dsDNA complex (Figure 3D) as in the case of the binary miR21–A1 incubate (Figure 3A). Obviously, in the presence of GNRs, the non-specific surface adsorption of free oligonucleotides (miR21 and A1) is unavoidable, sequestering a given fraction which during the incubation phase may not engage in the hybridization process. The fraction and nature of surface-adsorbed oligonucleotides is most probably dependent on their concentration, where a competitive displacement would favor the one found in a higher excess (e.g., 4-fold excess of miR21 vs. A1, Table 1). Upon the CGE analysis of the incubates containing GNRs, all surface-adsorbed oligonucleotides are to be detected in the electropherograms as free-moving ssDNA strands. As such, based on the levels of free-moving miR21 detected in the CGE analysis, a more detailed picture on the particular profile of concurrent specific (canonic) and non-specific interactions between the tested oligonucleotides and the gold surface may be drawn. Considering as a reference the theoretical 1:1 interaction, in an affinity-independent manner, between miR21 and a fixed level (2.22 μM) of one of its complementary oligonucleotides (A1 or A2), the fraction of canonically bound miR21 as a function of the added nominal concentration of miR21 (0.55–8.88 μM) would follow a decreasing trend (black line, Figure 4A).

In case of the binary incubates containing only free oligonucleotides (miR21–A1 and A2–miR21 systems), the experimental profile of hybridized miR21 (bound %miR21) demonstrates a superior affinity of A1 vs. A2 toward the target oligonucleotide. In the presence of GNRs (e.g., A2@GNRs–miR21), due to the concurrent non-specific surface adsorption of oligonucleotides, the decrease in specifically bound %miR21 is more abrupt, as the surface-adsorbed ssDNA during the incubation is detected as electrophoretically moving oligonucleotide in the CGE assays. As the adsorption of oligonucleotides on the gold surface does not imply significant conformational changes, it is most likely occurring at a higher rate than the hybridization process with the complementary oligonucleotides. This kinetic competition between the specific and non-specific interaction of the target with the oligonucleotide-functionalized gold nanostructures may also affect the binding affinity of the hybridization process, as the surface-adsorbed oligonucleotide layer (miR21 or its cognates) may sterically impede the formation of complementary oligonucleotide complexes. In the case of the ternary incubate (A2@GNRs–miR21–A1), the presence of the non-surface bound A1 ssDNA, with a superior affinity toward miR21 vs. free A2, is significantly altering the interaction profile upon increasing the concentration of target ssDNA. The preferential formation of the binary miR21–A1 complex seen by CGE translates into the rise of canonically bound %miR21 (gray column, Figure 4A) and explains the discrepancies observed for the ternary A2@GNRs–miR21–A1 vs. the binary A2@GNRs–miR21 system. The contribution of A1 in the interaction profile of the ternary vs. binary incubate is even more obvious when comparing the relative differences in the levels of free miR21 detected by CGE as the fraction of surface-bound miR21 induced by the GNRs should be constant (light blue column, Figure 4B). When comparing the relative differences between the surface-bound and free A2 binary incubates (A2@GNRs–mir21 vs. A2–mir21), a steady ~30% of free miR21 detected indicates the fraction of surface-adsorbed ssDNA being sequestered by the GNRs and preventing its specific interaction with the A2 oligonucleotide (dark blue, Figure 4B).

#### 2.1.2. Binding Stoichiometry in Binary ssDNA Interactions

To gain an even deeper knowledge on the affinity of hybridization of short ssDNA sequences with varying degrees of complementarity (i.e., miR21 and miR21-sp) and to assess the stoichiometry of their interaction with A1 or A2, the Job’s plot for the peak area recorded by CGE upon the incubation step was employed. The total concentration of the two interacting oligonucleotides, A1 and miR21, A2 and miR21 and A1 and miR21-sp, respectively, was kept constant at 26 μM while varying their molar ratio. The plot of the corrected electrophoretic peak area corresponding to the hybridized dsDNA complex recorded at 260 nm versus the molar ratio of miR21:A1 (Figure 5A), miR21-sp:A1 (Figure 5B) or miR21:A2 (Figure 5C) shows a clear maximum at a molar ratio of 1.16, 1.81 and 0.96, respectively (calculated based on the intersection of the two linear segments), indicating a 1:1 stoichiometry for the hybridized binary complex formed between miR21–A1 and miR21–A2. Nevertheless, in the case of miR21-sp, even if subtle changes in the slope indicate an (1:1) equimolar interaction, for the highest yield in hybridized complex formation, a molar ratio of 1.81 seems to be required.

#### 2.1.3. Binding Selectivity

To provide further information regarding the selectivity and predictability of base-pairing of A1 in binding miR21, three different types of ssDNA sequences with variable complementarity were investigated. A miR21-analogue having in the middle a non-interactive spacer of five thymine units (miR21-sp), one carrying complementary bases only toward the middle of the oligonucleotide sequence (anti-miR21) and a third sequence without any complementarity (random) toward miR21 (Figure 2A) were tested. Once again, binding studies pursued the same protocol, maintaining a fixed concentration of A1 (2.22 µM) and monitoring by CGE the hybridized dsDNA complex formation upon increasing concentrations of miR21-analogues. In the case of miR21-sp and A1 incubate, the detected hybridized dsDNA complex plateaued at a ~1.25 molar ratio, while A1 was completely consumed above a 1.5 molar ratio (Figure 6A). However, using oligonucleotide sequences with significantly lower complementarity, such as anti-miR21 or random, no dsDNA complex was identified, regardless of the used excess of miR21 analogue. As expected, the corrected electrophoretic peak area of miR21 analogues rose with the increase in their concentration (0–27.25 µM), whereas the corrected peak area of A1 remained steady (Figure 6B,C), clearly demonstrating a specific hybridization with miR21 and miR21-sp. Obviously, in a similar manner, the threshold of complementary nucleobases required to maintain or fine-tune the selectivity of a given oligonucleotide sequence for various analytical purposes may be achieved by rational engineering and fast CGE screening. Monitoring the molar ratio of miR21 analogue at which no electrophoretic peak for the free oligonucleotide (A1) is to be detected for the incubate in comparison with the same value for the reference nucleotide sequence (e.g., miR21) allows rough indications on the relative affinity between the probe (A1)–target oligonucleotide to be extracted (e.g., molar ratio >1 for miR21–A1 vs. >1.5 for miR21-sp–A1).

### 2.2. Microcalorimetric Study

Isothermal titration calorimetry is a fast and robust method to study the physical basis of molecular interactions, providing in a label-free manner all thermodynamic parameters of a molecular interaction after a single titration experiment [36]. Since beyond binding affinity, a thermodynamic characterization can provide comprehensive information about the driving force (enthalpic or/and entropic contribution) of the ligand association with its target [37], the ITC studies of our oligonucleotide systems were considered of vital importance. For all studied interactions, the binding reaction proved to be exothermic, and the heat profile was further integrated to obtain additional thermodynamic and affinity parameters related to the ssDNA hybridization process, such as reaction enthalpy (ΔH), change in entropy (∆S), binding stoichiometry (n) and dissociation constant (K_D_) (Table 2 and Table 3).

#### 2.2.1. Binary and Ternary Interactions

The study of binary systems (miR21–A1 or miR21–A2) (Figure 7A) revealed a higher affinity of miR21 toward A1, since the equilibrium dissociation constant (K_D_) of A1 toward miR21 is more than 3-fold lower in comparison with A2 to miR21 (K_D_ = 0.38 μM vs. K_D_ = 1.25 μM, respectively, *p* < 0.05). These data are also in line with the CGE observations (Figure 4A), and it is suspected of being related to the differences in the secondary structures of the two ssDNA oligonucleotides, where breaking the terminal hairpin loop of A2 being thermodynamically less favored for the initiation of the hybridization process with miR21 in comparison with the unpaired bases of the A1′s linear 3′ end (Appendix A). The overall free energy change of the interaction of miR21 with the two complementary ssDNAs favors the A1 oligonucleotide, with similar thermodynamic signatures to the binding energy (Figure 8A,B). Prior thermal denaturation of the ssDNA oligonucleotides proved to have no effect on the measured final affinity (Table 2). For both interacting binary systems, a 1:1 binding ratio has been observed in accordance with the CGE observations. To mimic the ternary interacting system studied by capillary electrophoresis, a mixture of A1 and A2 was titrated with miR21 (Figure 7B), demonstrating an affinity similar to that of A1 in its interaction with miR21 (K_D_ = 0.55 μM, *p* > 0.05) and an equimolar binding (n~1) between the three ssDNA sequences.

Further on, the ITC study of the interaction between A2@GNRs and miR21 revealed a higher reaction order (*n* = 3.18) and an even higher overall affinity (K_D_ = 0.022 μM), but in this case, the specific oligonucleotide–GNRs-linked oligonucleotide interaction remains confounded with the non-specific surface adsorption of the miR21 onto the gold surface (Figure 7C). Nevertheless, the recorded affinity was higher (K_D_ = 1.85 μM) with a 0.994 reaction order in the interaction between A2@GNRs with random (the ssDNA sequence with the lowest complementarity toward A2). Considering similar degrees of non-specific adsorption onto the gold surface of the investigated miR21 cognate analogues, the former ITC data once again proves, in an indirect manner, the existence of a specific interaction between A2 covalently attached to the surface of GNRs and miR21. Moreover, in the view of the ITC reaction orders of A2@GNRs against miR21 (*n* = 3.18) and random (*n* = 0.994), respectively, where the latter can only interact non-specifically with the A2-modified gold nanostructures, there is a clear indication of the 1/3rd contribution (~31.3%) of the metal surface adsorption to the overall specific oligonucleotide hybridization (dark blue, Figure 4B).

#### 2.2.2. Binding Selectivity

The intercalation of five thymine units as a non-interactive spacer into the middle of the miR21 strand, leading to a miR21 analogue (miR21-sp), is still enabling its hybridization with A1 yet again at a 1:1 binding ratio (Table 2). However, the equilibrium dissociation constant (K_D_) of A1 toward miR21-sp is one order of magnitude higher in comparison with miR21, demonstrating a much a stronger binding affinity between A1 and miR21. Furthermore, the thermodynamic signatures of the two tested oligonucleotides show very distinct enthalpic and entropic contributions to the binding energy. The overall free energy change upon miR21–A1 hybridization is dominated by a large binding enthalpy (ΔH = −201.23 kJ/mol) and a considerable unfavorable entropy (−TΔS = 164.467 kJ/mol) (Figure 8A). Most probably, the significant changes in the conformational, rotational, and translational freedom of the two ssDNA oligonucleotides upon binding is causing the high entropic penalty to the binding energy. In the case of miR21-sp, a much smaller binding enthalpy (ΔH = −18.3 kJ/mol) but a favorable entropic contribution (−TΔS = −10.3 kJ/mol) is noted. It is assumed that the non-interactive spacer of five thymine units in the miR21-analogue induces a higher degree of folding of the free miR21-sp in comparison with miR21, leading to lower penalties in ligand desolvation enthalpy and conformational entropy upon hybridization (Figure 8C). In accordance with the CGE results, once again, no hybridization event was detectable by ITC between A1 and random or anti-miR21 oligonucleotides (Figure 7D).

### 2.3. Molecular Dynamics Simulations

By using all-atom molecular dynamics approaches, the investigation aimed to provide additional and predictive biophysical insights and to establish which of the interacting systems are best suited for an appropriate recognition-binding mechanism.

The gyration measurements were used to establish the structural stability and the compactness levels of the modeled complexes. For the binary systems, the gyration profiles were stabilized after 10 ns of MD run, while implying significant structural deformations. As an overall structural behavior, as the number of nucleotides within the ssDNAs increases, the gyration profiles of the systems increase as well. The highest level of compactness was noted for miR21 when interacting with A2@GNRs (Appendix A).

The atomic root-mean-square-deviations (RMSD) were lower for A1 binary systems when compared to the equivalent A2@GNRs binary systems. RMSD for A1 presented the maximum average value of 0.64 nm in its interaction with the random oligonucleotide, while the minimum RMSD value of 0.49 nm was obtained for A1 interactions with miR21-sp. For A2@GNRs, RMSD values were comparable and presented the maximum average value of 1.13 nm in its interaction with anti-miR21, whereas the minimum RMSD average value of 1.06 nm was obtained for A2@GNRs interactions with miR21 oligonucleotide.

The Root-Mean-Square-Fluctuations (RMSF) profiles show lower atomic fluctuations for the binary interacting system A2@GNRs–miR21 (0.73 nm) and for the ternary system A2@GNRs–miR21–A1 (1.35 nm). Higher values were observed for the systems involving the other three oligonucleotides (miR21-sp, anti-miR21 and random) (Appendix A).

The H-bond plots show that A1 and A2@GNRs bind to miR21/miR21-sp/anti-miR21/random through strong hydrogen bond interactions. As expected, and in good agreement with the CGE and ITC data, the highest average number of H-bonds was noted for the interactions between A1/A2@GNRs with miR21 and miR21-sp oligonucleotides (within the 50 ns simulation frame ~27–30 with A1 and ~12–14 with A2@GNRs, respectively). In contrast, the lowest H-bond interactions were observed between A1–anti-miR21 and A1–random (~9 within the 50 ns simulation frame). The same observations emerged for the ternary systems as well, with the most abundant H-bonding observed in the miR21 and miR21-sp complexes, in contrast with the ones involving anti-miR21 and random (Figure 9).

The angle measurements revealed, as initially expected, similar average angles for the ternary systems involving miR21 and miR21-sp oligonucleotides (Appendix A–left). In good agreement with angle measurements, the solvent exposure profiles show similar hydrophobic rates for miR21 and miR21-sp oligonucleotides (Appendix A–right). The obtained total interaction energies are in good agreement with the forming H-bonds between the ssDNA oligonucleotides.

The visual inspection of the resulted MD trajectories (Figure 10) revealed a higher affinity between the ssDNAs based on the nucleotides’ complementarity for the interacting systems consisting of miR21 and miR21-sp structures. These findings are well correlated with the higher number of H-bonds between them, the highest total interaction energies and their correlated fluctuations extracted from PCA (Appendix A).

## 3. Materials and Methods

### 3.1. Materials

All chemicals used were of analytical grade purity. The oligonucleotides A1, A2, miR21, miR21-sp, random, and anti-miR21 (Figure 2A) were purchased from Integrated DNA Technologies IDT (Coralville, Iowa, USA). A2 oligonucleotide modified GNRs (A2@GNRs, 240 A2 units/rod) were purchased from Nanopartz Inc. (Loveland, CO, USA) under the Product no. C12L-10-850-OLIGO-DIH-50-1. The oligonucleotides were resuspended and stored according to the producer’s recommendations and used for experiments according to the specified validity period. The oligonucleotide sequences involved in the study and their possible interaction guided by complementarity are pictured in Figure 2A,B. OG, used as internal standard, was purchased from Sigma-Aldrich (St. Louis, MO, USA). The required value of pH was adjusted with 0.1 M NaOH or HCl solutions. The constituents of the samples and running buffer: *tris*(hydroxymethyl)aminomethane (TRIS), boric acid (BA), and ethylenediaminetetraacetic acid (EDTA) were obtained from Sigma-Aldrich (St. Louis, MO, USA), while dextran (2000 kDa) and glycerin were acquired from Pharmacosmos (Holbæk, Denmark). All buffers and working solutions were prepared in MilliQ water (18 MOhm). For all the oligonucleotide samples analyzed, a stock solution was prepared in 20 mM TBE (TRIS/Borate/EDTA) solution and further used to obtain the desired concentrations by subsequent dilutions with the same buffer. The concentration of A2 in the GNRs suspension (A2@GNRs) was calculated to be 5.57 µM, considering the technical information sheet provided by the manufacturer.

### 3.2. ssDNA Oligonucleotides Hybridization Conditions

For CGE studies, the specific oligonucleotide (A1, A2 or A2@GNRs, depending on the analyzed system) and target analyte were incubated off-capillary, at room temperature. As the distinct three-dimensional features (stems, loops, pseudoknots, hairpins, etc.) of the oligonucleotides maintained by hydrogen bonding, electrostatic and hydrophobic interactions (Appendix A) may induce particularities in their electrophoretic migration through the sieving media or may even influence the affinity toward their target, the effect of thermal denaturation (95 °C for 2 min, then rapid cooling on ice) before incubation and the effect of various physiological ions have also been investigated. In the case of binary complexes formed between ssDNA entities, experiments proved that hybridization happened within minutes after mixing the two corresponding ssDNA samples. In the case of ternary complexes, the hybridization was achieved by mixing the A2 ssDNA or A2@GNRs colloid with miR21 (the target oligonucleotide) and adding the corresponding third ssDNA strand involved in the supramolecular complex (A1) after 1 h. The reason behind this experimental design was to avoid competitive binding and to allow the annealing of miR21 to A2 ssDNA/A2@GNRs before having other non-surface bound strand with variable complementarity in solution.

### 3.3. Capillary Gel Electrophoresis (CGE)

The composition of the running buffer (RB) for gel electrophoresis was as follows: 8 mL 190 mM TBE buffer, 1 g of dextran and 1 g glycerin. Fused silica capillaries, with a 50 µm inner diameter and a total length of 50 cm (42 cm effective length), were purchased from Polymicro Technologies (R) (Phoenix, AZ, USA) and used throughout the studies. All CGE experiments were performed on an Agilent G1600 CE System equipped with a DAD detector, monitoring the separation of analytes at 260 nm. Each new capillary was first rinsed with 1M HCl for 3 min at 1 bar and ultrapure water for 5 min at 5 bars. Preconditioning of the capillary was completed by injecting ultrapure water for 5 min at 5 bars, HCl 1M for 3 min at 1 bar and RB for 10 min at 5 bars. Samples were injected hydrodynamically (104 mbar/sec) by the high-pressure option, whereas separation was performed at 25 °C using −30 kV. Signal recording and processing was performed by the 3D-CE ChemStation software (Agilent Technologies, Böblingen, Germany). The migration time corrected peak areas were normalized against the corrected peak area of OG, as internal standard (IS), if not otherwise specified.

### 3.4. Isothermal Titration Calorimetry

ITC was performed using an Affinity ITC microcalorimeter (TA Instruments, New Castle, DE, USA) controlled by its ITCRun v.3.6.5.0 Nano&Affinity ITC Data Collection software. All ITC analyses were conducted according to the following protocol unless otherwise specified. Prior to each measurement, each solution was degassed for 5 min under vacuum (25 inHg) at 25 °C. Then, the 300 μL oligonucleotide sample (the corresponding concentrations are presented in Table 2 and Table 3) was prepared in the following buffer: 10 mM TRIS, 100 mM NaCl, 0.1 mM EDTA, and pH = 8.5; afterwards, it was loaded in the ITC sample cell and titrated against the corresponding ssDNA analyte solution, prepared in the same buffer, at 25 °C. The reference cell of the calorimeter was filled with MiliQ water. Usually, 30 titration steps were performed, using a volume of 5 μL/step, except for the first and second injection, with 1 μL and 2 μL respectively. The thermodynamic parameters were obtained by fitting the titration curves against the build-in independent sites model using the NanoAnalyze v 3.11.0 software.

### 3.5. Computational Modeling of ssDNA Oligonucleotides Interacting System

An explicit TIP3P water model [38] was added to the following delineated (within vectors 5.83 × 5.05 × 25.99 Å3) systems: (1) a binary set of interacting structures between A1 and miR21/miR21-sp/anti-miR21/random, (2) another binary interacting set formed between A2 attached onto a gold surface (A2@GNRs) and miR21/miR21-sp/anti-miR21/random and (3) an interacting ternary assemble consisting of A1, A2@GNRs and miR21/miR21-sp/anti-miR21/random. The generated systems (Appendix A) were afterwards neutralized with Na(I) ions and subjected to 50,000 minimization steps using the steepest descend minimization algorithm [39] with defined position restraints on the last thymine residue of the A2 oligonucleotide situated at the gold surface. Prior to multiple unrestrained (except for the gold surface) MD productions for 50 ns, all systems were equilibrated in a canonical NVT ensemble for 100 ps at 310 K using the Nose–Hoover thermostat [40], a time constant of 2 ps and a single temperature coupling group. For the ssDNA complexes that involved interactions with A2 attached onto the gold surface, the non-equilibrium MD was carried out using a highly constraint AUI surface. All productions were performed using GROMACS package [41] and CHARMM27 force field [42]. The cut-off radius of the van der Waals potential was set to 1.1 nm, the long-range electrostatics were calculated using a particle mesh Ewald (PME) [43] implementation method and bonds involving hydrogen atoms were constrained using the holonomic LINCS algorithm [44].

## 4. Conclusions

The orthogonal but complementary electrophoretic and microcalorimetric data, assisted by molecular dynamics simulations offer a fast and complete mapping of the specific or non-specific, often competing, binary and higher order interactions in dynamic equilibria, occurring between various free and metal surface-bound oligonucleotides. The proof-of-concept study has been demonstrated on the DNA analogue of miR21 as a target molecule, as very often, such oligonucleotides are aimed to be specifically recognized and detected as biomarkers during affinity biosensor development. Critical affinity and selectivity attributes during different polyvalent interactions with other cognate oligonucleotides of variable complementarity (miR21-sp/anti-miR21/random) have been monitored with the proposed complementary set of analytical toolboxes. No interaction was observed between the A1 and A2 oligonucleotides and the non-complementary ssDNA sequences (anti-miR21/random) neither by ITC nor by CGE. The presence of a spacer between two complementary oligonucleotide sequences led to a decrease in the binary and ternary binding affinities. Additionally, subtle primary and secondary structure-dependent changes in the thermodynamic profile and in the corresponding affinity parameters were also detected, which often remain unnoticed and determine a suboptimal behavior of the biosensing platform.

The particular study of ternary interactions, complemented with computational modeling, is also able to quickly screen the impact and magnitude of the metal surface’s influence (non-specific adsorption, limited degrees of freedom upon covalent anchoring of the oligomeric strand) and the role of thermal denaturation on the hybridization event expected to occur between multiple ssDNA sequences with variable degrees of complementarity encountered in biosensing applications employing dual molecular recognition strategies.

## Figures and Tables

**Figure 1 ijms-24-00175-f001:**
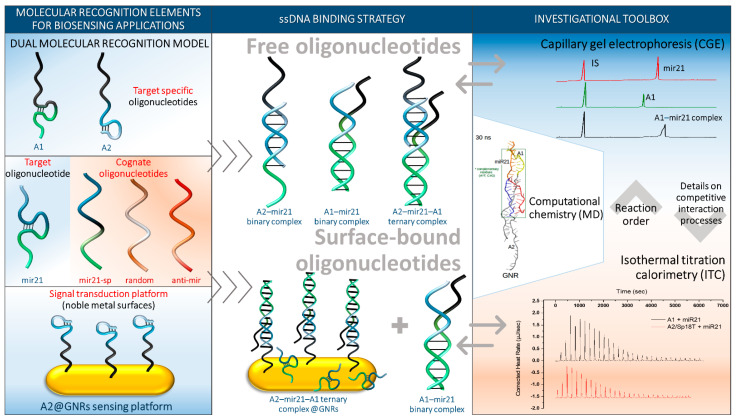
Schematic summary of the investigated scenarios and the employed analytical techniques.

**Figure 2 ijms-24-00175-f002:**
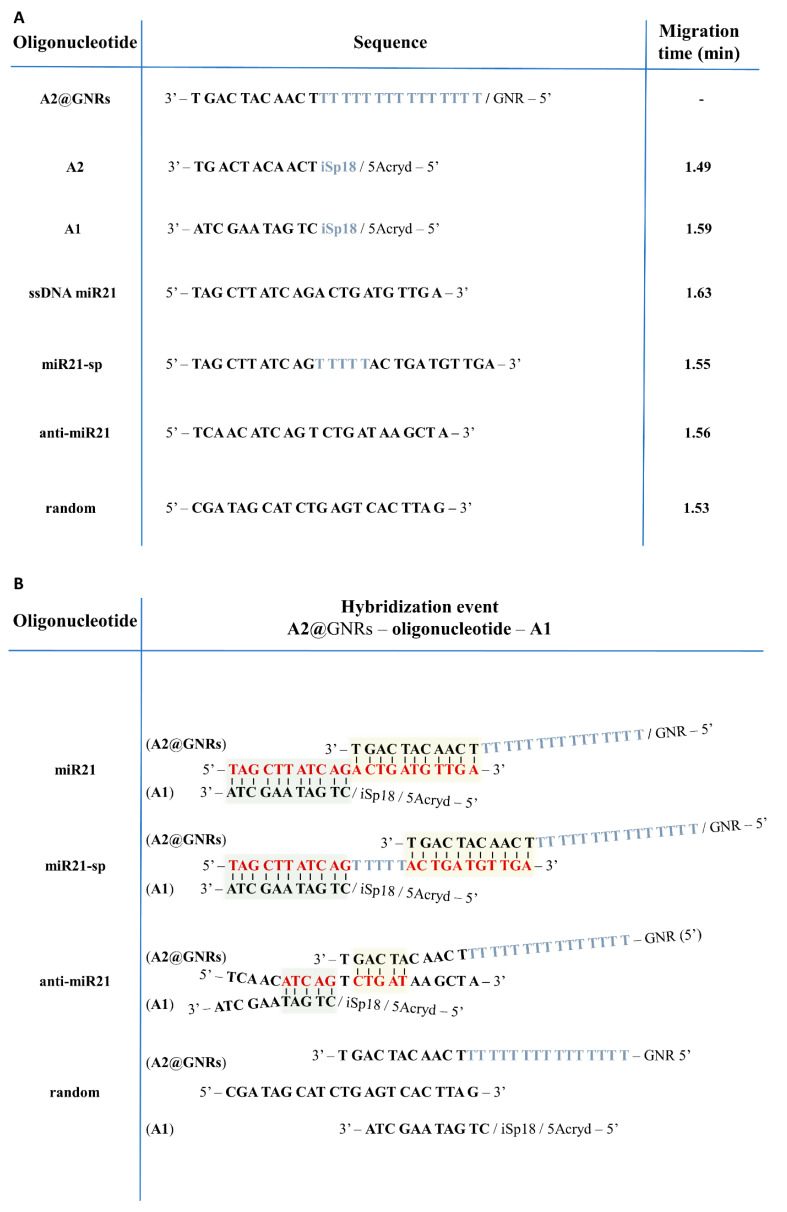
(**A**) The corresponding sequences and relative migration times of oligonucleotides involved in the study, calculated as relative to the migration time of the internal standard Orange G (OG), (**B**) Chemical structures and the expected complementarity of the DNA oligonucleotides involved in the study, The blue color has been used to representant the spacer oligonucleotides, the red color was used for the miR21 complementary oligonucleotides, light yellow and light green emphasize the hybridized regions of the two ssDNA strands.

**Figure 3 ijms-24-00175-f003:**
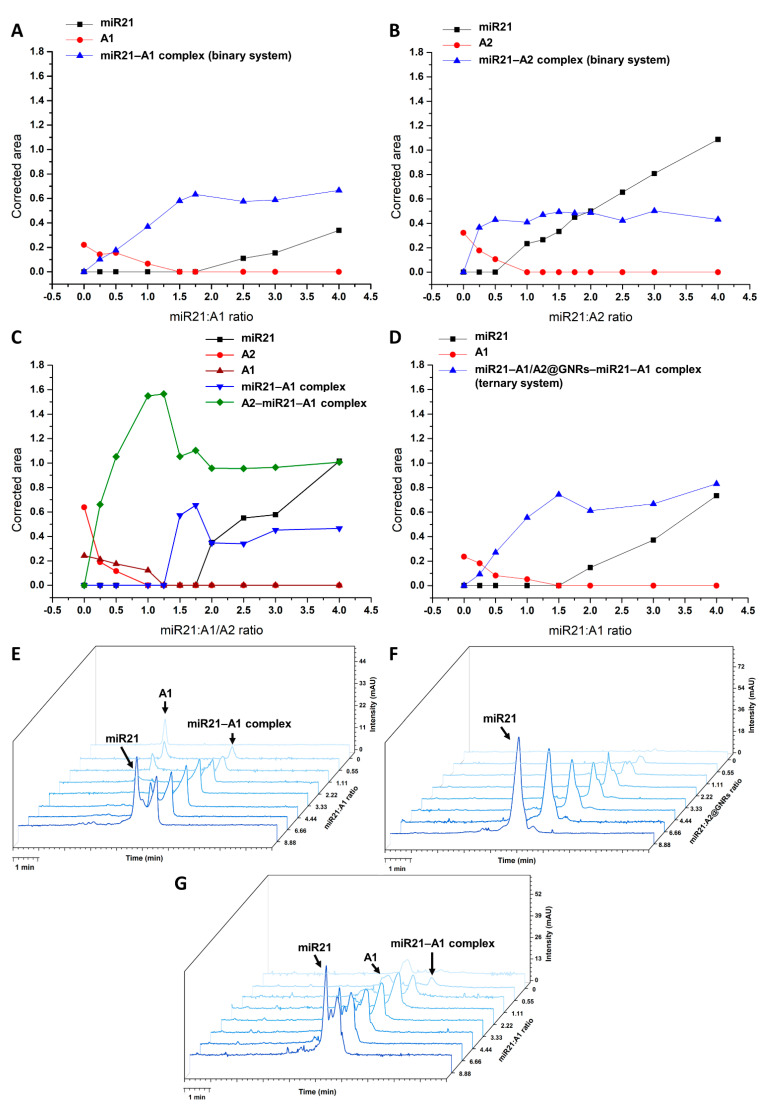
Binary system assessment between (**A**) miR21 (0–8.88 µM)–A1 (2.22 µM), (**B**) miR21 (0–8.88 µM)–A2 (2.22 µM), ternary system assessment between (**C**) A2 (2.22 µM)–miR21 (0–8.88 µM)–A1 (2.22 µM), (**D**) A2@GNRs (2.22 µM)–miR21 (0–8.88 µM)–A1 (2.22 µM) (obs: migration time corrected peak areas normalized against the corrected peak area of OG were used), corresponding waterfall plot for (**E**) miR21–A1 binary system, (**F**) A2@GNRs–miR21 binary system, (**G**) A2@GNRs–miR21–A1 ternary system.

**Figure 4 ijms-24-00175-f004:**
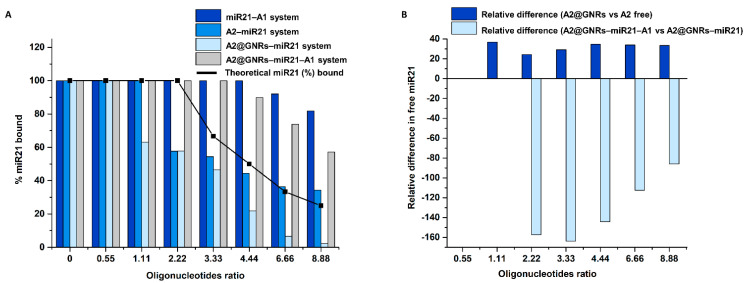
(**A**) Fraction of canonically bound miR21 to A1 and/or A2 as a function of the nominal concentration of miR21 added into various binary or ternary incubates studied, (**B**) Relative differences in the levels of free miR21 detected by CGE as a function of the nominal concentration of miR21 added, demonstrating the influence of GNRs and A1 on the interaction profiles of various binary or ternary incubates.

**Figure 5 ijms-24-00175-f005:**
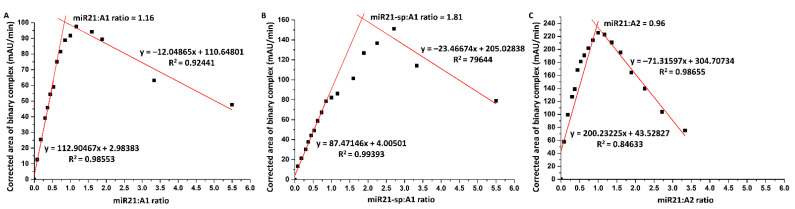
Job’s plot method for assessing stoichiometry for the interaction between (**A**) miR21–A1, (**B**) miR21-sp–A1, and (**C**) miR21–A2 (obs: migration time corrected peak areas were used).

**Figure 6 ijms-24-00175-f006:**
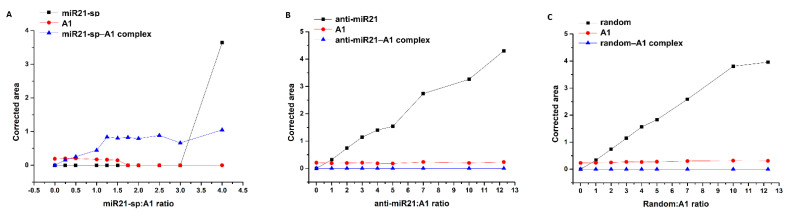
Assessment of miR21 analogues binding selectivity toward A1, using a fixed concentration of A1 (2.22 µM) and increasing concentration of the tested oligonucleotides: (**A**) miR21-sp (**B**) anti-miR21 (**C**) random (obs: migration time corrected peak areas normalized against the corrected peak area of OG were used).

**Figure 7 ijms-24-00175-f007:**
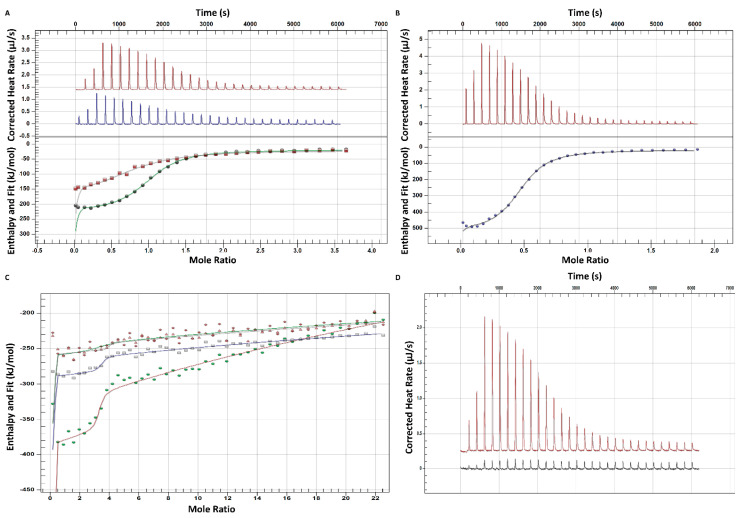
Integrated ITC data for the titration of (**A**) 10 µM A1 against 30 µM miR21 (red) and 10 µM A2 against 30 µM miR21 (blue), (**B**) 10 µM A1 + 10 µM A2 against 30 µM miR21, (**C**) Binding thermodynamic curve for the interaction between 0.97 µM A2@GNRs and 30 µM miR21 (green), 0.97 µM A2@GNRs and 30 µM miR21-sp (gray), 0.97 µM A2@GNRs and 30 µM random (red), (**D**) Corrected heat rate (µJ/s) plotted as a function of time for the ITC experiments studying the interaction between A1 and miR21 (red), A1 and anti-miR21 oligonucleotide (black).

**Figure 8 ijms-24-00175-f008:**
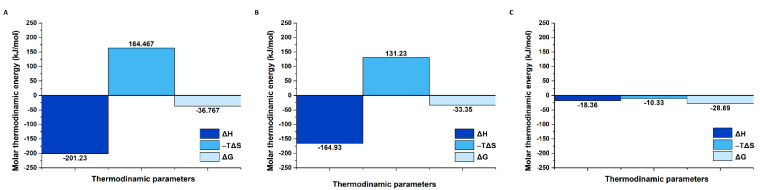
Thermodynamic data for the interaction between (**A**) A1 and miR21, (**B**) A2 and miR21, (**C**) A1 and miR21-sp.

**Figure 9 ijms-24-00175-f009:**
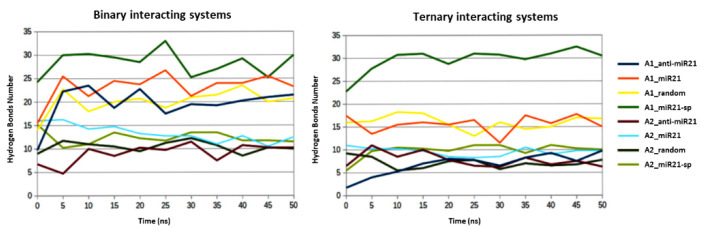
Hydrogen bond plots for A1/A2@GNRs and corresponding oligonucleotides.

**Figure 10 ijms-24-00175-f010:**
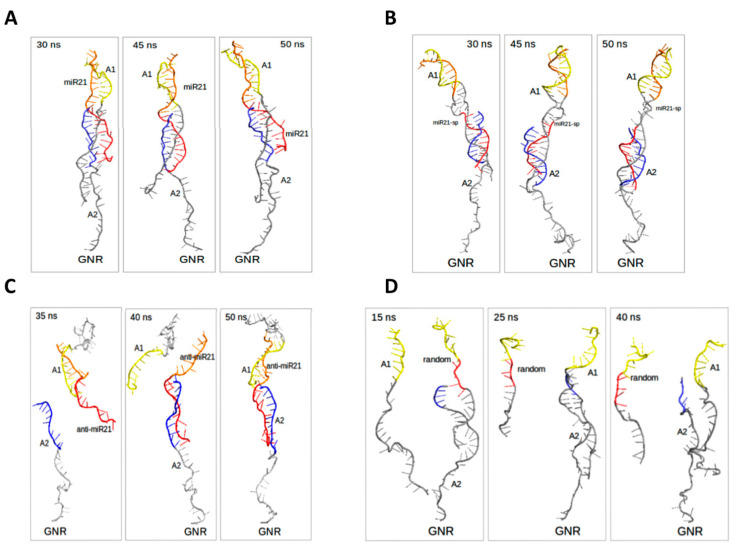
Resulted trajectories over 50 ns for miR21 (**A**) and miR21-sp (**B**) ternary systems and for anti-miR21 (**C**) and random (**D**) ternary systems.

**Table 1 ijms-24-00175-t001:** Free miR21 concentration (µM) detected for: binary system miR21–A1, A2–miR21, A2@GNRs–miR21 and ternary system A2@GNRs–miR21–A1.

Nominal (miR21) (µM)	Detected (Free) (miR21) (µM)			
	miR21–A1 System	A2–miR21 System	A2@GNRs–miR21 System	A2@GNRs–miR21–A1 System
0	0.00	0.00	0.00	0.00
0.55	0.00	0.00	0.00	0.00
1.11	0.00	0.00	0.41	0.00
2.22	0.00	0.94	1.48	0.00
3.33	0.00	1.52	2.49	0.00
4.44	0.00	2.47	4.01	0.45
6.66	0.53	4.23	6.50	1.74
8.88	1.61	5.83	8.81	3.80

**Table 2 ijms-24-00175-t002:** Thermodynamic parameters for the interaction of A1 with miR21 and its analogues and for the interaction between A2 and miR21.

	A1 10 µM + miR21 30 µM	A1 30 µM + miR21-sp 90 µM	A1 10 µM + Random/anti-miR21 30 µM	A2 10 µM + miR21 30 µM	A2 10 µM + miR21 30 µM *	(A1 10 µM + A2 10 µM) + miR21 30 µM
K_D_ (μM) (RSD%)	0.38 (38.25)	9.40 (24.56)	NB	1.25 (11.13)	1.55 (12.35)	0.55 (4.84)
N (RSD%)	1.118 (14.10)	0.973 (6.09)	-	0.96 (10.42)	1.018 (6.90)	0.45 (5.08)
ΔG (kJ/mol) (RSD%)	−36.767 (−2.93)	−28.69 (-2.57)	-	−33.7 (−0.79)	−33.17 (−0.91)	−35.70 (−0.33)
ΔH (kJ/mol) (RSD%)	−201.23 (−4.15)	−18.36 (26.01)	-	−164.93 (−11.39)	−174.9 (−4.86)	−508.5 (−1.45)
−TΔS (kJ/mol) (RSD%)	164.467 (5.72)	−10.33 (30.79)	-	131.23 (14.12)	141.76 (6.01)	472.76 (1.55)

NB—not binding. * A2 and miR21 thermally denatured.

**Table 3 ijms-24-00175-t003:** Thermodynamic parameters for the interaction of A2@GNRs with miR21 and its analogues.

	A2@GNRs 0.97 µM + miR21 30 µM	A2@GNRs 0.97 µM + miR21-sp 30 µM	A2@GNRs 0.97 µM + Random 30 µM
K_D_ (μM)	0.022	0.011	1.85
N	3.169	3.23	0.994
ΔH (kJ/mol)	−83.35	−37.85	−49.2
−TΔS (kJ/mol)	39.77	−6.279	16.6

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
