# Peer review of "Analytical Perspectives in the Study of Polyvalent Interactions of Free and Surface-Bound Oligonucleotides and Their Implications in Affinity Biosensing"

_ijms, 2022, doi:10.3390/ijms24010175_

Round 1
Reviewer 1 Report
The manuscript is well written, but it need some editing. References missing along the text. Figures need improving. Figure 1 is not clear; maybe it will be better to show the hybridisation in a separate figure; as it is very difficult to ready. All figure need and X and Y axis description; please complete the missing information, for example in figure 8. Most figures need improving and the style need to be similar to them, it appear the figures have been made by more then one person; so the style is clashing.
Reviewer 2 Report
Comments to author
The manuscript describe a complementary analytical workflow for the characterization of a series of free and surface-bound binary and ternary oligonucleotide interactions. Finding are interesting and can be considered for it publication after few important revision.
The specific comments are as:
1. A thorough English revision in suggested.
2. A scheme of whole work plan is highly recommended.
3. The presentation of data is very poor, graphs are barely visible, and this is something a serious concern.
4. How does the author confirmed the uniform assembly of these interaction over gold nanorods?
5. How about the stability of binary and ternary oligonucleotide on god surface?
6. How does these assembly behave in liquid environment, which is very important for sensor purposes?
